# Fractional SIR-Model for Estimating Transmission Dynamics of COVID-19 in India

**Nita H. Shah** [1,*], **Ankush H. Suthar** [1], **Ekta N. Jayswal** [1] **and Ankit Sikarwar** [2] 

1 Department of Mathematics, Gujarat University, Ahmedabad 380009, India; ankush.suthar1070@gmail.com (A.H.S.); jayswal.ekta1993@gmail.com (E.N.J.)
2 Department of Development Studies, International Institute for Population Sciences, Mumbai 400088, India; anks.sik@gmail.com
* Correspondence: nitahshah@gmail.com

**Abstract:** In this article, a time-dependent susceptible-infected-recovered (SIR) model is constructed to investigate the transmission rate of COVID-19 in various regions of India. The model included the fundamental parameters on which the transmission rate of the infection is dependent, like the population density, contact rate, recovery rate, and intensity of the infection in the respective region. Looking at the great diversity in different geographic locations in India, we determined to calculate the basic reproduction number for all Indian districts based on the COVID-19 data till 7 July 2020. By preparing district-wise spatial distribution maps with the help of ArcGIS 10.2, the model was employed to show the effect of complete lockdown on the transmission rate of the COVID-19 infection in Indian districts. Moreover, with the model's transformation to the fractional ordered dynamical system, we found that the nature of the proposed SIR model is different for the different order of the systems. The sensitivity analysis of the basic reproduction number is done graphically which forecasts the change in the transmission rate of COVID-19 infection with change in different parameters. In the numerical simulation section, oscillations and variations in the model compartments are shown for two different situations, with and without lockdown.

**Keywords:** Indian districts; COVID-19; population density; the basic reproduction number; SIR-model; numerical simulation

## Article Highlights

- A time-dependent susceptible-infected-recovered (SIR) model is constructed which includes the most effective parameters related to the intensity of the infection, population density and contact rate.
- Looking at the great diversity and differences in different geographic locations in India, the value of the basic reproduction number is calculated based on the COVID-19 data from Indian districts which forecasts the transmission rate of COVID-19 infection in different situations.
- We found that the performance of the proposed SIR model is different for different fractional-order nonlinear dynamical systems.
- District-wise spatial distribution maps illustrating reproduction rates with and without lockdown measures are prepared with the help of ArcGIS 10.2.

### MSC 37NXX

## 1. Introduction

Humans have been battling against the outbreak of infectious diseases throughout their history. In the food market of Wuhan city of China, the infection of a new severe acute respiratory syndrome coronavirus 2 (COVID-19) primarily appeared in December 2019 [1]. While the disease was strange to many and highly contagious, people are disappointed to stop its transmission throughout the world [2]. It rapidly turned into the largest coronavirus

outbreak. The outbreak is still going on, moreover, the ongoing situation reflects that during 2021, the mortality rate of the infection is going to be even higher than last year. Investigation using population models says that at the initial stage of the COVID-19 outbreak, the basic reproduction number, which is the average number of secondary infections caused by one infected individual during the incubation period fell between 1.5 and 3.5 globally [3]. Until now, the disease rapidly spread in almost all countries, and the global number of COVID-19 cases is rising at an accelerated rate. As of 22 April 2021, 144,213,316 cases were confirmed worldwide causing 3,065,499 deaths (worldometer.com). After the USA, India has borne the brunt of the epidemic, reporting 15,924,732 cases (worldometer.com). To counter the epidemic circumstances, effective steps are demanded like complete lockdown, sanitizing infected areas, and developing the extent of medical facilities. The transmission of COVID-19 has been investigated by experts from different disciplines to overcome the epidemic situation.

Following a voluntary public curfew in India, on 24 March 2020, the Government of India (GOI) commanded a nationwide complete lockdown in the country for 21 days as a preventive measure against the COVID-19 pandemic [4]. The lockdown was extended to a few months longer by local and state governments based on the severity of the infection. This lockdown slowed the growth rate of the pandemic to a rate of doubling of infected cases every eight days (R0 close to 2) [5]. Then, sequentially the GOI has implemented unlocking strategies followed by the enforcement of a series of regulations in the infected regions [6,7].

Indian districts have much geographic, social, and economic diversity; hence the transmission rate of COVID-19 was altered in each region. It is observed that the transmission of COVID-19 is extremely affected by the contact rate and density of the infected individuals in the region. It is also seen in a previous study that the contact rate is directly proportional to the population density of the region [8]. Hence, the transmission capacity of infection was even high in areas with high population densities like West Bengal, Delhi, Maharashtra, Bihar, and Ahmedabad. Considering these observations, we have made a basic compartmental model that analyzes the transmission rate using parameters based on population density and contact rate. To understand the spread of COVID-19 in a large area, the investigation of the spread of disease should be done in each possibly smallest distinctive region. Hence, we were inspired to calculate the numerical value of the transmission rate of the disease at the district level in India.

In the fight against the COVID-19 outbreak, the use of mathematical modeling improves our knowledge of disease dispersion and preventive measures. The formation of an efficient epidemiological model is a challenging task. The uncertainty of the transmission dynamics can be divided and allocated to different sources of uncertainty in the concerned parameters. Gupta et al. used long-term climatic records of various geographical parameters, air temperature, rainfall, evapotranspiration, solar radiation, humidity, wind speed, and population density at the regional level to investigate the density of COVID-19 infections. Moreover, their study suggests that the relatively hot and dry regions in lower altitudes of the Indian Territory are more prone to the COVID-19 infection [9]. Rafiq et al. developed a prognostic yet deterministic model by identification techniques to forecast the spread of COVID-19 for 30 days in the ten most affected states of India [10]. Using linear mixed models with random intercept and fixed slopes, Sy, Karla et al. has defined the association between population density (used as a proxy for contact rates) and the basic reproduction number of COVID-19 across the U.S. The statistical analysis done by Sy et al., concludes that regions with greater population density have greater rates of transmission of COVID-19 [11]. Mahajan et al. constructed a SIPHERD-model to analyze the impact of lockdown and the number of tests conducted per day on COVID-19 transmission and then predicted the total number of confirmed, active, and death cases [12]. Under the fractional-order derivative, Shaikh et al. has formulated a compartmental model and applied potential control strategies during the COVID-19 outbreak to estimate the effectiveness of preventive measures [13].

Many compartmental models have increased our knowledge of COVID-19 propagation evaluating under the assistance of geographical parameters related to the disease [14]. Roda et al. has examined that the predictions using more complex models may not be more substantial compared to a simpler model [15]. Therefore, for smooth functioning, we choose to make an SIR model based on the COVID-19 outbreak in India. The model includes parameters associated with population density and contact rate which helps to predict the transmission of COVID-19 in various geographically dense areas of India. Correlation of the population density with contact rate helps to investigate the basic reproduction number before and after lockdown. Furthermore, the graphical analysis of the SIR model and the spatial distribution of reproduction rates in Indian districts are performed with and without lockdown.

## 2. Mathematical Modelling

### 2.1. Formulation of the SIR-Model

In this section, a compartmental SIR model is formulated. The model contains three compartments, susceptible individuals ($S$), infected individuals ($I$), and recovered individuals ($R$). Recovered individuals from COVID-19 gained sufficient protective immunity to the virus so that the possibility of reinfection time extends to at least six months [16]. Hence, in the model, we have assumed that individuals in the recovered state gain total immunity to the virus.

In our models of population dynamics, variation in the compartment due to the natural birth rate ($B$) and natural mortality rate ($\mu_2$) are assumed to be time-independent. The density of infected individuals ($\delta$) in the contaminated region is given by multiplying population density constant ($C$) to the ratio of the total number of infected individuals by COVID-19 to the total population of the same region. The contact rate ($\beta$) is an important parameter as the susceptible individuals certainly get infected by the virus when they come in contact with the infected individuals. The recovery rate ($\gamma$) is the ratio of the recovered individuals to the infected individuals; and the mortality rate ($\mu_1$) of COVID-19 infection is the ratio of the total number of dead individuals due to COVID-19 to the infected individuals. The SIR-model is given by the following system of equations.

$$
\begin{aligned}
S' &= B - (1+\delta)\beta SI - \mu_2 S \\
I' &= (1+\delta)\beta SI - \gamma I - \mu_1 I \\
R' &= \gamma I - \mu_2 R
\end{aligned}
\tag{1}
$$

In the SIR-model, $\beta SI$ shows the quantity of newly emerged cases of COVID-19 through direct contact between an infected individual and a susceptible individual. $\delta\beta SI$ shows the regional intensity of newly emerging infection through direct contact with infected individuals as the transmission of COVID-19 depends on the intensity of infection in the region. The natural death rate of the infected individuals ($\mu_2 I$) during the incubation period of the infection is negligible, hence it is ignored in the SIR-model.

### 2.2. The Feasible Region

We know that $(S(t), I(t), R(t)) \geq 0$ if $(S(0), I(0), R(0)) \geq 0$. Based on system (1), we have $N = B - \mu_2 N + (\mu_2 - \mu_1)I$, which implies $(1 + \mu_2)N = B + (\mu_2 - \mu_1)I$. Where, $N = S + I + R$, and when $t \to \infty$, we have $N \leq \frac{B}{1+\mu_2}$. Hence $N$ is bounded and the feasible region ($\Lambda$) for the system (1) is as follows:

$$
\Lambda = \left\{ (S, I, R) \in R_+^3 / S + I + R \leq \frac{B}{1 + \mu_2} \right\}
\tag{2}
$$

### 2.3. Equilibrium Points and the Basic Reproduction Number

The solutions of system (1) are called equilibrium points. The SIR-model has two equilibrium points as follows:

1. Disease free equilibrium point: $E_0 = \left( \frac{B}{\mu_2}, 0, 0 \right)$
2. Endemic equilibrium point:

$$E^* = \left( \frac{\gamma + \mu_1}{(1+\delta)\beta}, \frac{B\beta(1+\delta) - \mu_2(\gamma + \mu_1)}{\beta(1+\delta)(\gamma + \mu_1)}, \frac{\gamma(B\beta(1+\delta) - \mu_2(\gamma + \mu_1))}{\beta\mu_2(1+\delta)(\gamma + \mu_1)} \right)$$

The threshold value or the basic reproduction number $(R_0)$ of COVID-19, is formulated using the next-generation matrix algorithm [17,18]. $R_0$ is defined as the average number of secondary infected cases rising from an average primary case in an entirely susceptible population.

The dynamical system (1) is split into two disjoint matrices, $f$ and $v$. $F$ and $V$ are the Jacobian matrices of matrices $f$ and $v$ respectively, where the matrix $F$ shows the new infectious rates and the matrix $V$ shows other rate of infection transferred in between the compartments.

$$F = \begin{bmatrix} \frac{(1+\delta)\beta B}{\mu_2} & 0 & 0 \\ 0 & 0 & 0 \\ 0 & 0 & 0 \end{bmatrix}, V = \begin{bmatrix} \gamma + \mu_1 & 0 & 0 \\ \frac{(1+\delta)\beta B}{\mu_2} & \mu_2 & 0 \\ -\gamma & 0 & \mu_2 \end{bmatrix}, \text{ and } FV^{-1} = \begin{bmatrix} \frac{(1+\delta)\beta B}{\mu_2(\gamma + \mu_1)} & 0 & 0 \\ 0 & 0 & 0 \\ 0 & 0 & 0 \end{bmatrix}$$

The basic reproduction number for the system (1) is obtained as the spectral radius of the matrix $(FV^{-1})$ around the disease-free equilibrium point. i.e.,

$$\rho(FV^{-1}) = \frac{(1+\delta)\beta B}{\mu_2(\gamma + \mu_1)} \tag{3}$$

**Definition 1.** *The Caputo fractional order derivative of a function $y$ in the interval $[0, T]$ is defined by,*

$$^C D_{0+}^\alpha y(t) = \frac{1}{\Gamma(n-\alpha)} \int_0^t (t-s)^{n-\alpha-1} y^{(n)}(s) \, ds \tag{4}$$

*where, C represents the Caputo derivative, $D^\alpha$ denotes the Caputo fractional derivative of order $n = [\alpha] + 1$ and $[\alpha]$ represents the integer part of $\alpha$.*

**Definition 2.** *Laplace transform of the Caputo derivative is defined as,*

$$L\{D^\alpha y(t)\} = s^\alpha y(s) - \sum_{k=0}^{n-1} s^{\alpha-k-1} y^{(k)}(0), \ n-1 < \alpha < n, \ n \in N. \tag{5}$$

*After applying Caputo derivative to system (1) one can have [19],*

$$\begin{aligned} ^C D^\alpha S &= B - (1+\delta)\beta SI - \mu_2 S \\ ^C D^\alpha I &= (1+\delta)\beta SI - \gamma I - \mu_1 I \\ ^C D^\alpha R &= \gamma I - \mu_2 R \end{aligned} \tag{6}$$

Here, $C$ denotes the Caputo derivative having order $\alpha \in (0, 1]$ while $B, \beta, \gamma, \delta, \mu_1$, and $\mu_2$ are positive parameters as given in model (1) with initial conditions $S(0) = S_0, I(0) = I_0, R(0) = R_0$. The system Equation (6) is solved using the Laplace–Adomian decomposition

method. The system of differential Equation (6) is converted into a system of algebraic equations using Laplace transformation.

$$
\begin{aligned}
L\{^{C}D^{\alpha}S(t)\} &= L\{B - (1+\delta)\beta SI - \mu_2 S\} \\
L\{^{C}D^{\alpha}I(t)\} &= L\{(1+\delta)\beta SI - \gamma I - \mu_1 I\} \quad \text{or} \\
L\{^{C}D^{\alpha}R(t)\} &= L\{\gamma I - \mu_2 R\}
\end{aligned}
$$

$$
\begin{aligned}
s^{\alpha}L\{\text{ or }S(t)\} - s^{\alpha-1}S(0) &= L\{B - (1+\delta)\beta SI - \mu_2 S\} \\
s^{\alpha}L\{I(t)\} - s^{\alpha-1}I(0) &= L\{(1+\delta)\beta SI - \gamma I - \mu_1 I\} \\
s^{\alpha}L\{R(t)\} - s^{\alpha-1}R(0) &= L\{\gamma I - \mu_2 R\}
\end{aligned}
\tag{7}
$$

From the above Expressions, Equation (7), we obtain the form

$$
\begin{aligned}
L\{S(t)\} &= \frac{S_0}{s} + \left[\frac{1}{s^{\alpha}}L\{B - (1+\delta)\beta SI - \mu_2 S\}\right] \\
L\{I(t)\} &= \frac{I_0}{s} + \left[\frac{1}{s^{\alpha}}L\{(1+\delta)\beta SI - \gamma I - \mu_1 I\}\right] \\
L\{R(t)\} &= \frac{R_0}{s} + \left[\frac{1}{s^{\alpha}}L\{\gamma I - \mu_2 R\}\right]
\end{aligned}
\tag{8}
$$

Then, the above form Equation (8) is obtained in the form of series by assuming solutions $S(t)$, $I(t)$ and $R(t)$ in the form of infinite series. Moreover, nonlinear term $S(t)I(t)$ is also decomposed as follows:

$$
S(t) = \sum_{n=1}^{\infty} S_n(t), \; I(t) = \sum_{n=1}^{\infty} I_n(t), \; R(t) = \sum_{n=1}^{\infty} R_n(t) \text{ and } S(t)I(t) = \sum_{n=1}^{\infty} A_n(t) \tag{9}
$$

where, $A_n$ is an Adomian polynomials which is defined as follows:

$$
A_n = \frac{1}{\Gamma(n+1)} \frac{d^n}{dK^n} \left[\sum_{i=0}^{n} K^i S_i(t) \sum_{i=0}^{n} K^i I_i(t)\right]\Bigg|_{K=0} \tag{10}
$$

Substituting Equation (9) in Equation (8), we get:

$$
\begin{aligned}
L\left\{\sum_{n=1}^{\infty} S_n(t)\right\} &= \frac{S_0}{s} + \left[\frac{1}{s^{\alpha}}L\left\{B - (1+\delta)\beta \sum_{n=1}^{\infty} A_n(t) - \mu_2 \sum_{n=1}^{\infty} S_n(t)\right\}\right] \\
L\left\{\sum_{n=1}^{\infty} I_n(t)\right\} &= \frac{I_0}{s} + \left[\frac{1}{s^{\alpha}}L\left\{(1+\delta)\beta \sum_{n=1}^{\infty} A_n(t) - \gamma \sum_{n=1}^{\infty} I_n(t) - \mu_1 \sum_{n=1}^{\infty} I_n(t)\right\}\right] \\
L\left\{\sum_{n=1}^{\infty} R_n(t)\right\} &= \frac{R_0}{s} + \left[\frac{1}{s^{\alpha}}L\left\{\gamma \sum_{n=1}^{\infty} I_n(t) - \mu_2 \sum_{n=1}^{\infty} R_n(t)\right\}\right]
\end{aligned}
\tag{11}
$$

Matching both sides in each equation in (11) with iterative algorithm, we get:

$$
\begin{aligned}
L\{S_{n+1}\} &= \frac{B}{s^{\alpha}} - \frac{(1+\delta)\beta}{s^{\alpha}}L\{A_n\} - \frac{\mu_2}{s^{\alpha}}L\{S\} \\
L\{I_{n+1}\} &= \frac{(1+\delta)\beta}{s^{\alpha}}L\{A_n\} - \frac{\gamma}{s^{\alpha}}L\{I\} - \frac{\mu_1}{s^{\alpha}}L\{I\} \\
L\{R_{n+1}\} &= \frac{\gamma}{s^{\alpha}}L\{I\} - \frac{\mu_2}{s^{\alpha}}L\{R\}
\end{aligned}
\tag{12}
$$

Taking the Laplace inverse in each equation of (12), we have general form:

$$
\begin{aligned}
S(n+1) &= S(n) + \frac{t^{\alpha}}{\Gamma(\alpha+1)}(B - (1+\delta)\beta SI - \mu_2 S) \\
I(n+1) &= I(n) + \frac{t^{\alpha}}{\Gamma(\alpha+1)}((1+\delta)\beta SI - \gamma I - \mu_1 I) \\
R(n+1) &= R(n) + \frac{t^{\alpha}}{\Gamma(\alpha+1)}(\gamma I - \mu_2 R)
\end{aligned}
\tag{13}
$$

Here, $t$ defines time. To study the mathematical behavior corresponding to the equilibrium of system (6), a different value of $\alpha$ is used.

### 3. Data-Driven Forecasting of Covid-19 in India

We found limited literature associating the regional evaluation of COVID-19 transmission in India. Hence, for the average elevation of the spread of COVID-19 in the possibly smallest distinctive regions of India, we have calculated the transmission rate of the disease at the district level in India.

It is observed that restriction on contact rate is an effective step in reducing transmission of COVID-19. The population density constant ($C$) is used to calculate the density of infected individuals ($\delta$), which is hypothetically taken as 500 near to the average population density of India. To govern the contact rate in a particular region, lockdown is a relevant action. Hence, the growing transmission intensity of the virus requires a complete lockdown. But some essential activities are necessary to manage medical, food, and research facilities to survive in this COVID-19 outbreak. To fulfil this thought, the contact rate ($\beta$) during the complete lockdown condition is taken hypothetically as 0.3, and without lockdown as 0.6.

The total population and its growth rate for Indian states and districts are taken from the 15th National Census Survey conducted by the Census Organization of India [https://www.census2011.co.in/district.php], [https://www.citypopulation.de/php/india--admin.php?adm2id=769]. The natural death rate for the world is taken 0.00396 [https://www.worldometers.info/] and for India, it is taken 0.073 [https://www.cia.gov/library/publications/the-world-factbook/geos/in.html]. The district-wise total number of infected, recovered and death cases due to COVID-19 up to 7 July 2020 was taken from [https://www.grainmart.in/news/covid--19--coronavirus--india--state--and--district--wise--tally/]. All the above data was accessed on 7 July 2020. (Table 1).

The purpose of the SIR-model is to overview the spread of COVID-19 in India at the ground level with and without lockdown conditions. By practicing all the foregoing data, the value $R_0$ is calculated for the districts of India. To implement the lockdown situation in the SIR-model, the value of the contact rate is taken low ($\beta = 0.3$) and without lockdown, its value is taken high ($\beta = 0.6$).

Figures 1 and 2 present the spatial distribution of reproduction rate ($R_0$) in Indian districts with and without lockdown respectively. It is found that the $R_0$ has increased significantly without lockdown measures in almost all the districts of India. In the majority of the districts, its values are greater than 2, which shows the critical condition of the spread of COVID-19 infection in several regions of India.

**Table 1.** Parametric values for COVID-19 in the world, India and Indian states founded on 7 July 2020.

| Place | $\mu_1$ | B | $\delta$ | $\gamma$ |
|---|---|---|---|---|
| World | 0.04491 | 0.00944 | 0.79691 | 0.58339 |
| India | 0.02713 | 0.177 | 0.32977 | 0.62355 |
| Maharashtra | 0.04258 | 0.1764 | 0.94322 | 0.54372 |
| Tamil Nadu | 0.01366 | 0.1561 | 0.79683 | 0.57899 |
| Delhi | 0.0309 | 0.2121 | 3.00906 | 0.715 |
| Gujarat | 0.0532 | 0.1917 | 0.3052 | 0.71417 |
| Uttar Pradesh | 0.02825 | 0.202 | 0.07166 | 0.66731 |
| Rajasthan | 0.02228 | 0.2131 | 0.1509 | 0.78683 |
| West Bengal | 0.03389 | 0.1384 | 0.12592 | 0.66277 |
| Madhya Pradesh | 0.04037 | 0.2035 | 0.10522 | 0.75759 |
| Haryana | 0.01577 | 0.199 | 0.3452 | 0.76183 |
| Karnataka | 0.01588 | 0.2716 | 0.20707 | 0.41589 |
| Andhra Pradesh | 0.01194 | 0.1098 | 0.11834 | 0.44558 |
| Bihar | 0.00799 | 0.2542 | 0.05831 | 0.7425 |
| Telangana | 0.01189 | 0.1358 | 0.36463 | 0.5744 |
| Jammu and Kashmir | 0.01591 | 0.2364 | 0.34586 | 0.61303 |
| Assam | 0.00119 | 0.1693 | 0.18828 | 0.67164 |
| Odisha | 0.00504 | 0.1758 | 0.15281 | 0.68087 |
| Punjab | 0.02604 | 0.1389 | 0.11698 | 0.69234 |
| Kerala | 0.00498 | 0.176 | 0.08416 | 0.59417 |
| Uttarakhand | 0.01329 | 0.1881 | 0.1567 | 0.8181 |
| Chhattisgarh | 0.00424 | 0.1806 | 0.05134 | 0.8 |
| Jharkhand | 0.00701 | 0.2242 | 0.04326 | 0.7246 |
| Tripura | 0.00059 | 0.1484 | 0.23027 | 0.72045 |
| Ladakh | 0.001 | 0.1387 | 3.76441 | 0.83184 |
| Goa | 0.00386 | 0.0823 | 0.62151 | 0.58522 |
| Himachal Pradesh | 0.00929 | 0.1753 | 0.07845 | 0.69638 |
| Manipur | 0 | 0.245 | 0.24336 | 0.52806 |
| Manipur | 0 | 0.245 | 0.24336 | 0.52806 |
| Chandigarh | 0.01232 | 0.1719 | 0.23071 | 0.82341 |
| Puducherry | 0.01385 | 0.2873 | 0.53194 | 0.47478 |
| Nagaland | 0 | 0.6441 | 0.15795 | 0.3888 |
| Mizoram | 0 | 0.2348 | 0.08977 | 0.67513 |
| Arunachal Pradesh | 0.00741 | 0.2603 | 0.09756 | 0.34074 |
| Sikkim | 0 | 0.1289 | 0.10236 | 0.52 |
| Dadra and Nagar Haveli and Daman and Diu | 0 | 0.5588 | 0.58189 | 0.4375 |
| Andaman and Nicobar Islands | 0 | 0.0686 | 0.18524 | 0.52482 |
| Meghalaya | 0.01136 | 0.2795 | 0.01483 | 0.48864 |

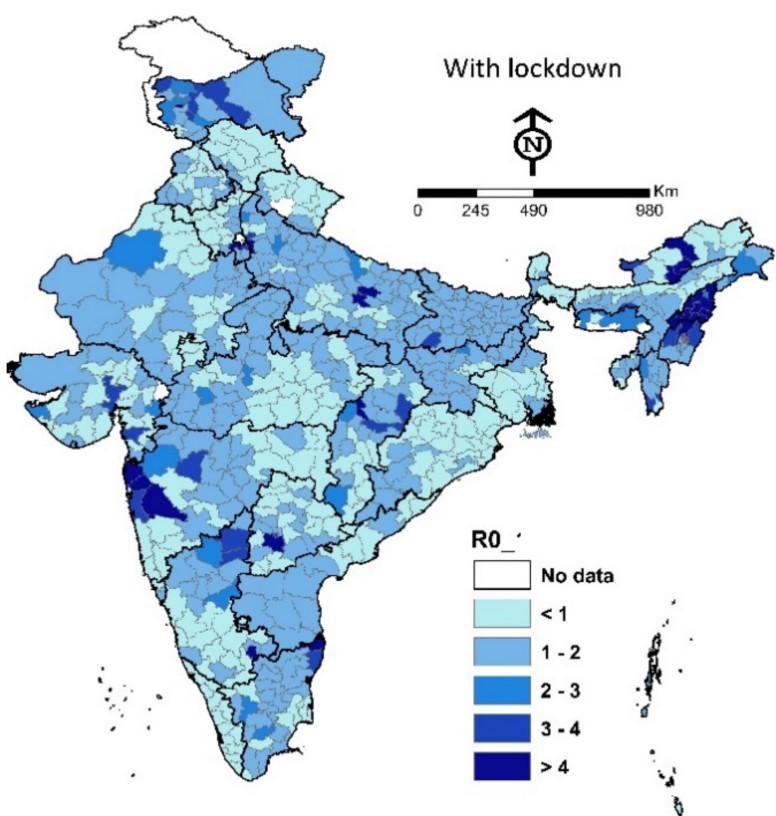

**Figure 1.** Regional transmission of COVID-19 during the lockdown.

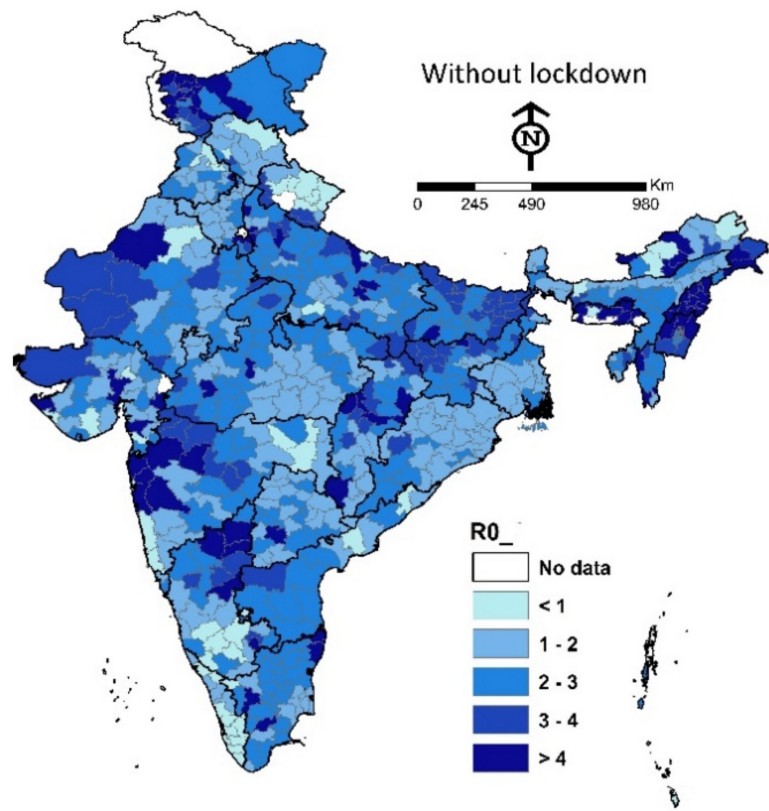

**Figure 2.** Regional transmission of COVID-19 without lockdown.

## 4. Susceptibility of $R_0$ with Respect to the Parameters

To evaluate the intensity of the effect of parameters on the transmission rate of COVID-19, $R_0$ is plotted under the variation in the parameters used in the SIR-model. The parametric value ($B = 0.177$, $\beta = 0.3$, $\gamma = 0.6235$, $\mu_2 = 0.073$ and $\mu_1 = 0.0271$) used in the sensitivity is based on the data of the COVID-19 outbreak in India going on 7 July 2020.

Change in the basic reproduction number ($R_0$) with respect to the parameters, contact rate ($\beta$), recovery rate ($\gamma$), mortality rate of the COVID-19 ($\mu_1$), and regional intensity of the COVID-19 ($\delta$) is plotted in Figures 3–6 respectively. In Figure 3, it is observed that in a complete lockdown situation ($\beta \leq 0.3$), $R_0$ is less than 1.4. As we know, there is an exponential rise in the number of infection cases with time when $R_0$ exceeds 1. During this COVID-19 outbreak, $R_0$ is consistently estimated as greater than 2 in most of regions in India. Figure 6 shows that the strict lockdown curfew in India can reduce the $R_0$ by up to 1.12. The intensity of infection in India is high enough that even lockdown is not enough to control the spread of the COVID-19 virus. Figure 3, Figure 4, and Figure 6 explain that the $R_0$ is less than one when the contact rate ($\beta$) is less than 0.20, or the recovery rate ($\gamma$) is more than 0.94, or the mortality rate of COVID-19 ($\mu_1$) greater than 0.34.

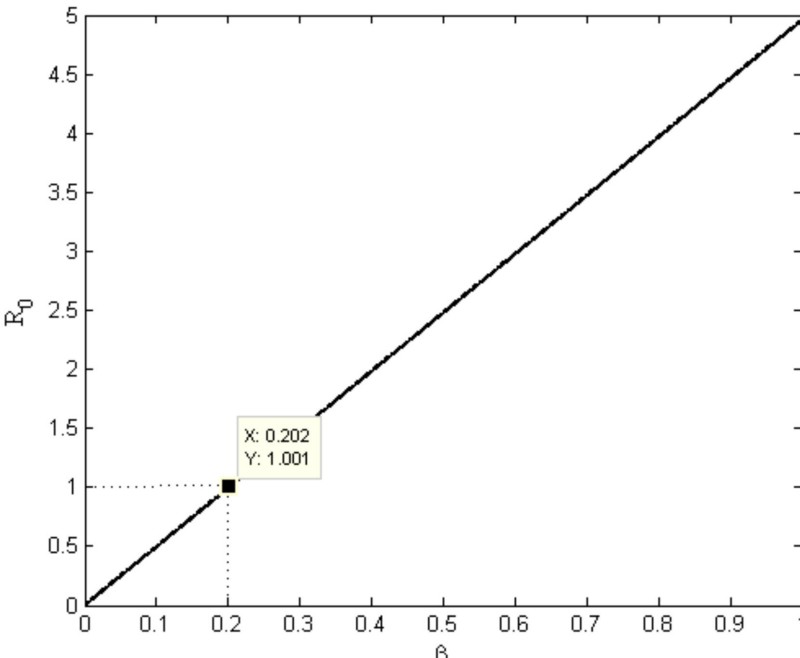

**Figure 3.** Change in $R_0$ with respect to the contact rate.

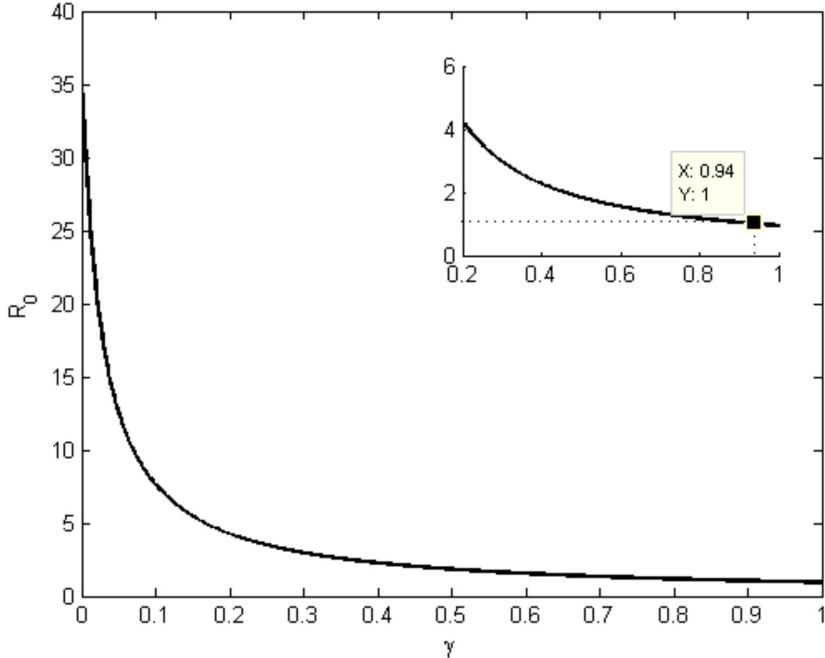

**Figure 4.** Change in $R_0$ w.r.t the recovery.

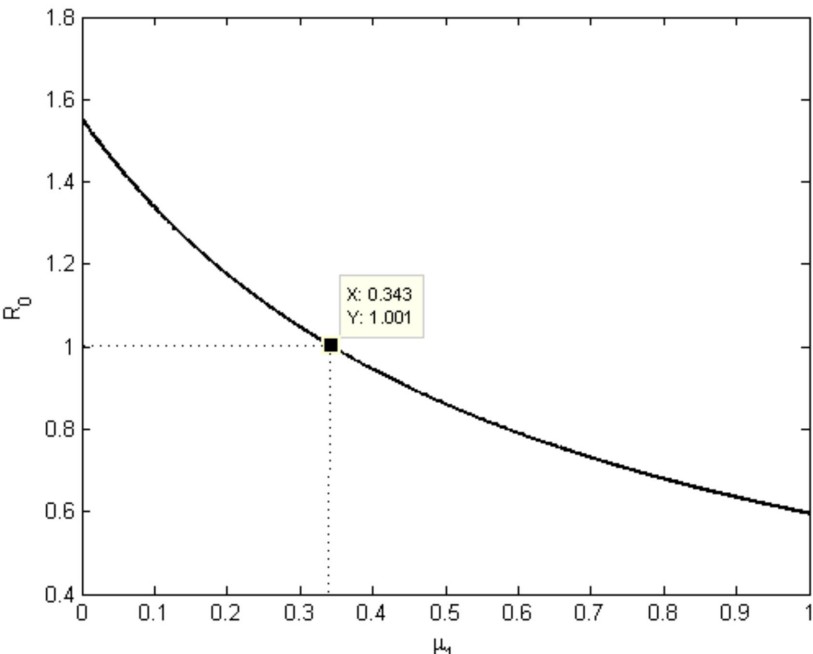

**Figure 5.** Change in $R_0$ w.r.t the mortality rate of COVID-19.

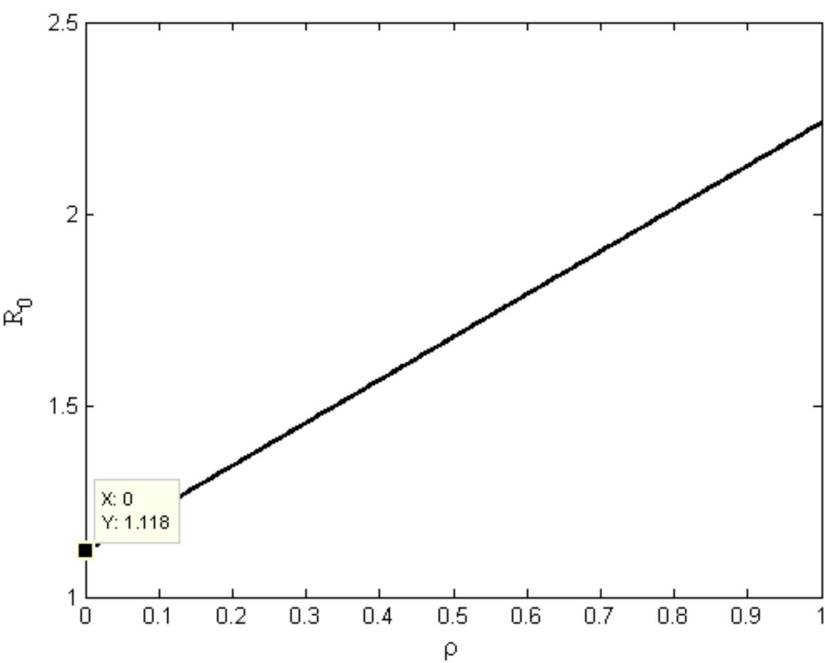

**Figure 6.** Change in $R_0$ w.r.t the regional intensity of the COVID-19.

## 5. Numerical Simulation

Figure 7a,b shows the variation in each compartment of the model with time w.r.t COVID-19 cases in the world and India respectively. The values of the parameters for Figure 7a are $B = 0.0094$, $\beta = 0.3$, $\gamma = 0.5839$, $\mu_1 = 0.0449$, $\mu_2 = 0.0039$, and $\delta = 0.7969$; the values of the parameters for Figure 7b are $B = 0.177$, $\beta = 0.3$, $\gamma = 0.6235$, $\mu_1 = 0.0271$, $\mu_2 = 0.0730$, and $\delta = 0.3297$. The initialization of susceptible, infected, and recovered compartments is given by $S(0) = 0.3$, $I(0) = 0.8$, and $R(0) = 0.2$ respectively. Evaluation of recovery rates of COVID-19 in the world and India indicate that the recovery rate in India is high.

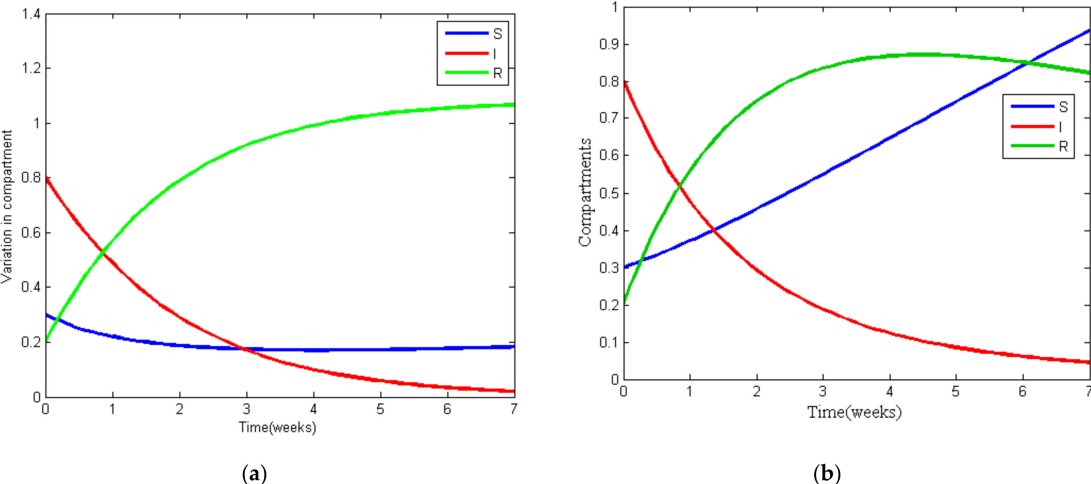

(a) (b)

**Figure 7.** (**a**) Variation in the SIR-model for data of COVID-19 cases in the world. (**b**) Variation in the SIR-model for data of COVID-19 cases in India.

The approximate solutions $S(t)$, $I(t)$, and $R(t)$ under lockdown situations ($\beta = 0.3$) in India are displayed in Figure 8a–c, respectively. In each figure, three different values of the fractional order $\alpha$, $\alpha = 0.6$, 0.8, and 1 are considered.

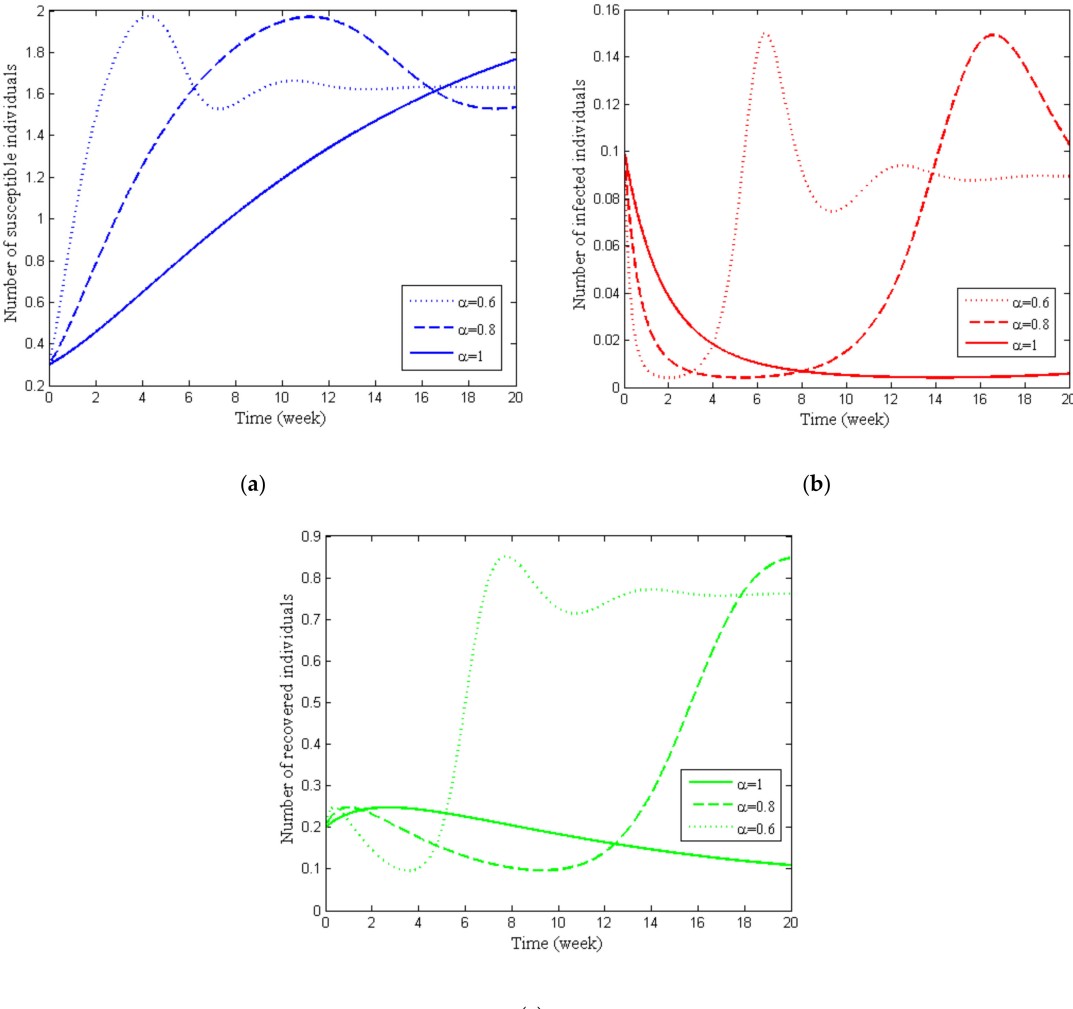

**Figure 8.** (**a**) Variation in susceptible class with change in $\alpha$, (**b**) Variation in infected class with change in $\alpha$, (**c**) Variation in recovered class with change in $\alpha$.

Figure 9a,b displays oscillations in the susceptible, infected, and recovered compartments due to the COVID-19 outbreak in the world; Figure 10a,b also displays the oscillations in the compartments due to the outbreak in India. Figures 9a and 10a describe the without lockdown situation ($\beta = 0.6$); and Figures 9b and 10b describe with the situation with lockdown ($\beta = 0.3$). It can be observed that the oscillations in Figure 9a (Figure 10a) are denser than the oscillations in Figure 9b (Figure 10a); it expresses that the intensity of the outbreak is higher without a lockdown situation.

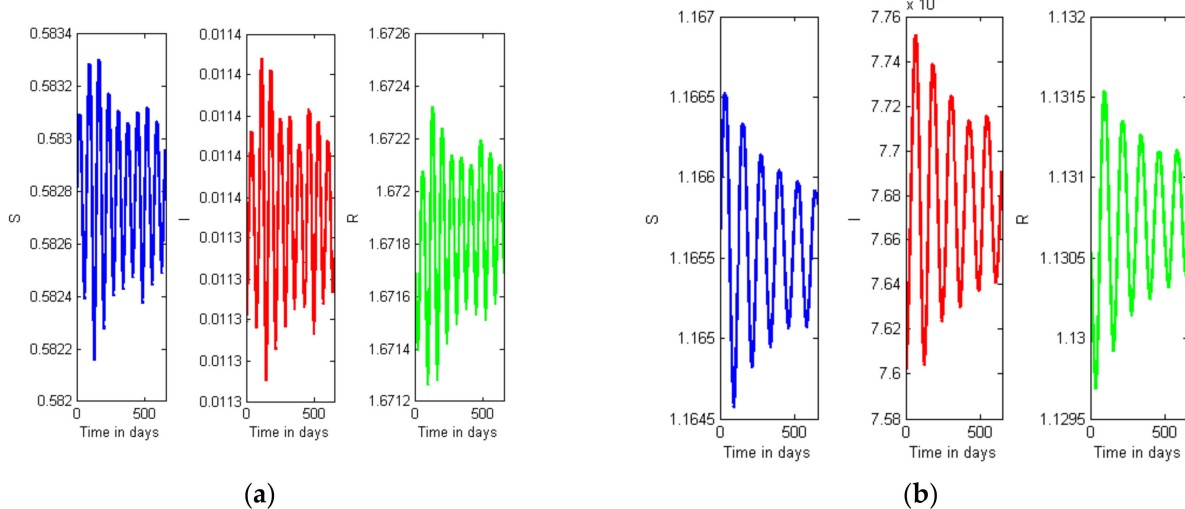

**Figure 9.** (**a**) Oscillations in the compartments without lockdown (World). (**b**) Oscillations in the compartments with lockdown (World).

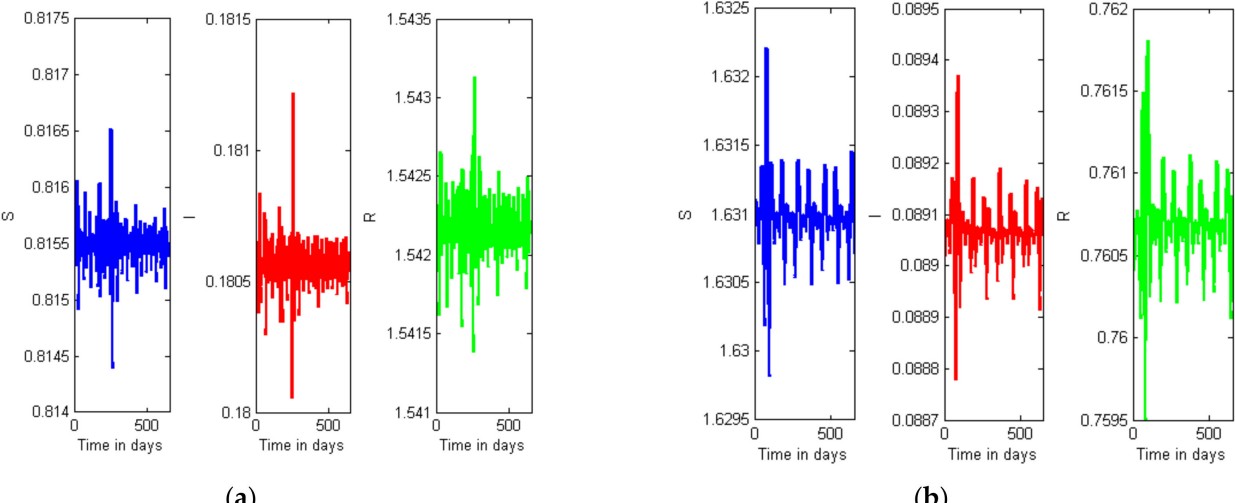

**Figure 10.** (**a**) Oscillations in the compartments without lockdown (India). (**b**) Oscillations in the compartments with lockdown (India).

## 6. Discussion and Conclusions

Looking at the current epidemic situation due to the COVID-19 outbreak, a theoretical and pathological investigation is urgently needed to control the transmission of COVID-19. The intention of forming the SIR model is to evaluate the spread of COVID-19 in the possibly smallest distinctive regions of India and to analyze nature of the transmission with respect to the fundamental parameters like population density, contact rate, recovery rate and intensity of infection in the various regions.

Since lockdown is directly affected by the contact rate, the transmission rate analysis was done for two different values of contact rate, $\alpha = 0.3$ when lockdown is implemented and $\alpha = 0.6$ without lockdown. The patterns of spatial distribution of the basic reproduction number $(R_0)$ in Indian districts are demonstrated in Figures 1 and 2 for with and without lockdown situations, respectively. It is found that the $R_0$ has grown significantly without the lockdown strategy across all districts of India. In Figure 9a,b and Figure 10a,b, the positive impact of lockdown is observed in the form of the density of oscillations of the compartments.

Variation in the basic reproduction number ($R_0$) concerning the parameters of the SIR-model is shown the Section 4. This sensitive analysis of $R_0$ shows that after contact rate the most effective parameter to the transmission rate is the intensity of infection in the respective region. Hence the transmission rate of COVID-19 is observed high in the urbanized districts where population density is higher, like Mumbai, Delhi, Ahmedabad, etc. The analysis suggests that with only a strict lockdown, we cannot control the spread of infection effectively. Besides, locking down for a long time is also not suggested due to economic sustainability. Hence, with a partial lockdown, other effective control strategies are suggested to reduce the density of infection in highly affected districts.

A notable reason for the rapid spread of the novel virus is the contact rate of asymptomatically infected individuals, who are mostly asymptomatic yet contagious for a long incubation period. One could extend the work by adding a parameter for the contact rate of symptomatic and asymptomatic separately to analyze the transmission of COVID-19 more intensely.

**Author Contributions:** Conceptualization, N.H.S. and A.H.S.; Data curation, A.H.S., E.N.J. and A.S.; Formal analysis, A.H.S. and A.S.; Investigation, N.H.S.; Methodology, N.H.S.; Resources, A.H.S. and A.S.; Software, E.N.J. and A.H.S.; Supervision, N.H.S.; Visualization, N.H.S., E.N.J. and A.S.; Writing—Original draft, A.H.S. and E.N.J.; Writing—Review & editing, A.H.S. All authors have read and agreed to the published version of the manuscript.

**Funding:** All the authors are thankful to DST-FIST file # MSI-097 for technical support to the Department of Mathematics, Gujarat University. The second author (AHS) is funded by a Junior Research Fellowship from the Council of Scientific & Industrial Research (file no.-09/070(0061)/2018-EMR-I). Third author (ENJ) is funded by UGC granted National Fellowship for Other Backward Classes (NFO-2018-19-OBC-GUJ-71790).

**Acknowledgments:** Authors thank anonymous reviewers for their constructive comments.

**Conflicts of Interest:** Authors do not have any conflict of interest.

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
