# Peer review of "Fractional SIR-Model for Estimating Transmission Dynamics of COVID-19 in India"

_2571-8800, doi:10.3390/j4020008_

Round 1
Reviewer 1 Report
The paper represents a study of transmission rate for Covid-19 in India based on the time-dependent modification of SIR epidemic model. The authors proposed a formalization of the model, calibration of model parameters, according to statistical data of new Covid cases in different regions of India. They also represent the sensitivity analysis of the basic reproductive number graphically in different pandemic scenarios.
The positive points of the research include an interesting framework, formalization model and scenario analysis of the reproductive ratio coefficient, informative literature review. However, there exists some comments and remarks:
Unfortunately, the modification of the SIR model is not very new, but it allows analysis of an epidemic situation in the considered framework. Probably it would be interesting to add an asymptomatic subgroup into consideration.
It is not clear what time interval is considered in the paper to perform the analysis of the transmission of COVID-19.
It would be recommended to add details in the system’s transformation on p. 7, probably it can be added into the Appendix section. Please define the parameter in formula (5).
Did you compare the solutions received by your modified SIR model and real statistical data of spreading COVID-19 in India?
Which software did you use to perform the analysis of the virus spreading?
Please add captions and enlarge figures in lines 213-214 pp. 12-13 to improve its readability. Probably it would be more readable if you can change the green color in fig.8(c)
Author Response
#Reviewer 1
- Unfortunately, the modification of the SIR model is not very new, but it allows analysis of an epidemic situation in the considered framework. Probably it would be interesting to add an asymptomatic subgroup into consideration.
Reply: Yes, the research can be expanded by adding an asymptomatic compartment. But it is hard to get real district-wise data of asymptomatically infected individuals in India. Moreover, we come to know from one study that accuracy in the outcome of the SIR model is more compared to other complicated population models. Hence, to get the accurate outcome from real data of COVID-19 outbreak we choose the SIR model. Thank you.
- It is not clear what time interval is considered in the paper to perform the analysis of the transmission of COVID-19.
Reply: The simulation is performed for data taken on 7th, July 2020. We have predicted the upcoming transmission rate of the infection in Indian districts using the preceding data till 7th July 2020. Moreover, change in variables is considered per week which is mentioned in simulation part. Thank you.
- It would be recommended to add details in the system’s transformation on p. 7, probably it can be added into the Appendix section. Please define the parameter in formula (5).
Reply: A detailed calculation of the system’s transformation is added as per recommendation. Also, parameters in a fractional dynamical system are described. Thank you.
- Did you compare the solutions received by your modified SIR model and real statistical data of spreading COVID-19 in India?
Reply: By observing the transmission pattern of the infection influenced by population density in India, our study of the threshold value is foretold high in populated Indian districts without a lockdown situation which is matched to the current situation in India. Also, it is shown in sensitive analysis that the contact rate and the regional intensity of the infection are highly affecting the transmission rate of the infection which is one of the major reasons for the current increasing intensity of COVID-19 in India. Thank You.
- Which software did you use to perform the analysis of the virus spreading?
Reply: The simulation part is done using MATLAB software and district-wise distribution maps are prepared with the help of ArcGIS 10.2. Thank You.
- Please add captions and enlarge figures in lines 213-214 pp. 12-13 to improve its readability. Probably it would be more readable if you can change the green color in fig.8(c).
Reply: Captions are added and figures are enlarged as per suggestions. To show the simulation for the class of recovered individuals we have used green color throughout the article. Similarly, blue for susceptible class and red for infected class. Hence, we have used green color in figure 8(c) to keep color monotony. Still, we can change color if needed. Thank you.
Reviewer 2 Report
The paper “Fractional SIR-Model for Estimating Transmission Dynamics of COVID-19 in India” investigated transmission rate of COVID-19 in India at the fundamental level, a time-dependent susceptible-infected-recovered (SIR) model. To improve the quality, the following recommendations can be incorporated.
Before all, the structure of this paper is similar to the technical report, not an academic paper, so authors should again rewrite all of parts base on journal paper style. Especially, introduction and abstract. Sections Funding, Conflicts of interest/Competing interests, Availability of data and material, Code availability, Authors' contributions, and … should move to end of the manuscript
1.The authors should review the other investigation on their study way in the introduction part and finally note the novelty of the article. The introduction part needs to develop. The font size of the Introduction section should be same.
- The methodology section is not well organized for the readers to understand the concept.
- please describe your validation of models.
- The language used in the introduction can be more specific to the scope and aim of the study.
- pages 5, 6, and 7 for each equation should add a number.
- Section “Conclusions” is poorly written and limited. More details on quantities should be provided. In fact, the main body of your manuscript should be a result, not backgrounds and methodologies.
7- the quality of Figure 9, and 10 is very poor.
Author Response
#reviewer 2
- Before all, the structure of this paper is similar to the technical report, not an academic paper, so authors should again rewrite all of parts base on journal paper style. Especially, introduction and abstract. Sections Funding, Conflicts of interest/Competing interests, Availability of data and material, Code availability, Authors' contributions, and … should move to end of the manuscript.
Reply: We have tried to modify the introduction and abstract part as per the guidance. Sections Funding, Conflicts of interest…etc., these technical information are moved to end of the manuscript. Thank you.
- The authors should review the other investigation on their study way in the introduction part and finally note the novelty of the article. The introduction part needs to develop. The font size of the Introduction section should be same.
Reply: Modify the introduction part as per the requirement. Font in the introduction section is resized. Thank you.
- The methodology section is not well organized for the readers to understand the concept.
Reply: A detailed calculation of the system’s transformation to fractional dynamical system and parametric description is added to understand method easily. Thank you.
- please describe your validation of models.
Reply: By observing the transmission pattern of the infection influenced by population density in India, our study of the threshold value is foretold high in populated Indian districts without a lockdown situation which is matched to the current situation in India. Also, it is shown in sensitive analysis that the contact rate and the regional intensity of the infection are highly affecting the transmission rate of the infection which is one of the major reasons for the current increasing intensity of COVID-19 in India. Thank You.
- The language used in the introduction can be more specific to the scope and aim of the study.
Reply: We have included the scope and aim of the study in the introduction part. Thank you.
- pages 5, 6, and 7 for each equation should add a number.
Reply: The number is added to the respective equations. Thank You.
- Section “Conclusions” is poorly written and limited. More details on quantities should be provided. In fact, the main body of your manuscript should be a result, not backgrounds and methodologies.
Reply: The conclusion section is modified. Thank you.
- the quality of Figure 9, and 10 is very poor.
Reply: Respective figures are enlarged to improve its visibility. Thank you.
Round 2
Reviewer 1 Report
Firstly, I would like to say thank you for the authors for revised version.
Now it is necessary to check for typos in the article. I have found a couple of typos, please find my comments in the attachment.

Author Response
Corrected typos as suggested. Thank you for reviewing the manuscript.
Reviewer 2 Report
Since the information in this article may be helpful to researchers in the field of COVID-19, I am interested in accepting this article.
Author Response
Thank you for your positive response.